# Knee Cartilage Change within 5 Years after Aclr Using Hamstring Tendons with Preserved Tibial-Insertion: A Prospective Randomized Controlled Study Based on Magnetic Resonance Imaging

**DOI:** 10.3390/jcm11206157

**Published:** 2022-10-19

**Authors:** Yuhan Zhang, Shaohua Liu, Yaying Sun, Yuxue Xie, Jiwu Chen

**Affiliations:** 1Department of Orthopaedics, Beijing Chaoyang Hospital, Capital Medical University, Beijing 100020, China; 2Department of Sports Medicine, Huashan Hospital, Fudan University, Shanghai 200040, China; 3Department of Radiology & Institute of Medical Functional and Molecular Imaging, Huashan Hospital, Fudan University, Shanghai 200040, China; 4Department of Sports Medicine, Shanghai General Hospital, Shanghai Jiaotong University, Shanghai 200080, China

**Keywords:** ACLR, hamstring tendon with preserved tibial insertion, MRI, T2, cartilage volume

## Abstract

**Background:** Comparing to anterior cruciate ligament reconstructions (ACLR) with free hamstring tendon (FHT), ACLR with preserved tibial-insertion hamstring tendon (HT-PTI) could ensure the blood supply of the graft and avoid graft necrosis. Yet, whether HT-PTI could protect the cartilage and clinical outcomes in mid-long period after ACLR was still unclear. **Purpose:** To compare the cartilage change and clinical results between the HT-PTI and FHT in 5 years after ACLR. **Study design:** Randomized controlled trial; Level of evidence, 2. **Methods:** A total of 45 patients who underwent isolated ACLR with the autograft of hamstring tendons were enrolled and randomized into 2 groups. The study group undertook ACLR with HT-PTI, whereas the control group had FHT. At pre-operation, and 6, 12, 24, and 60 months post-operation, all cases underwent evaluation with Knee Injury and Osteoarthritis Outcome Score (KOOS), and MR examination. The knee cartilage was divided into 8 sub-regions of which the T2 value and cartilage volume on MRI were measured and documented. The data of two groups were compared and their correlations were analyzed. **Results:** A total of 18 patients in the HT-PTI group and 19 patients in the FHT group completed the follow-up. The KOOS scores were improved at each follow-up time point (*p* < 0.001), reached the most superior at 12 months and maintained until 60 months but had no significant difference between the two groups. At 60 months, the cartilage in most subregions in FHT group had higher T2 values than those of pre-operation (*p* < 0.05) and also higher than HT-PTI group; The cartilage volume changes (CV%) are positive at 6 months and negative from 12 to 60 months in the FHT group, while being negative at all time points in the HT-PTI group. The values of absolute CV% in most subregions in FHT group were significantly higher than those in the HT-PTI group at 6 and 60 months (*p* < 0.05). **Conclusion:** The improvement of KOOS score peaked at 12 months in all cases and had no difference between the two groups. The cartilage in the FHT group had more volume loss, earlier and wider damage than that in the HT-PTI group within 5 years. No significant correlation was found among KOOS score, CV%, and T2 value.

## 1. Introduction

The instability of the knee joint after the injury of the anterior cruciate ligament (ACL) will lead to the wear of articular cartilage. At present, relevant research shows that ACL reconstruction (ACLR) can effectively correct the instability of the knee joint, thereby reducing the wear of articular cartilage [1]. However, various studies showed that the damage of articular cartilage was still progressing after ACLR had corrected joint instability [2,3,4,5,6]. It was proved that various inflammatory factors including IL-1β, IL-6, IL-10 and TNF-α were released into the joint during the graft necrosis and proliferation stages after ACLR, activated matrix metalloproteinases to digest collagen and proteoglycan in cartilage matrix, and resulted in cartilage degeneration [7,8]. Therefore, the potential biochemical and metabolic factors could lead to the occurrence of knee OA after ACLR [1,9].

In the early stage of cartilage degeneration, the cartilage would have increased water content, decreased proteoglycan content [10] and reduced volume [11,12]. MRI is the most commonly used to evaluate cartilage injury [12,13,14] because its sensitivity in detecting the water change in cartilage [15]. As the T2 relaxation time of cartilage is directly proportional to the water distribution in cartilage and inversely proportional to the specific distribution of proteoglycan [12,16,17], MRI quantitative T2 value is used for detecting early cartilage lesions through the changes of cartilage matrix and water content [18].

The growing activity in the field of cartilage damage creates a need for validated clinical outcome scores whose special emphasis was given to patients with cartilage injuries [19]. Different from the Lysholm, Tegner, and international knee documentation committee (IKDC) scores, the Western Ontario and McMaster Universities Index (WOMAC) is commonly used to measure the patients with osteoarthritis (OA) [20]. However, the population presenting with focal cartilage lesions after ACLR is generally younger and more active as compared to patients with OA [21]. The Knee Injury and Osteoarthritis Outcome Score (KOOS) was developed as an extension of the WOMAC and designed to assess symptoms and function in younger or more active patients with ACL injuries and cartilage damage [19]. Therefore, the KOOS would fit this population better. In addition, the KOOS have been proved to be a measure of sufficient reliability, validity, and responsiveness for surgery and physical therapy after ACLR [19,22].

Given the hamstring tendon with intact tibial insertion (HT-PTI) had much less necrosis than the free hamstring tendon (FHT) after ACLR [7,23], and could avoid necrosis and reduced the level of intra-articular inflammation [7,23,24], we hypothesized that the knee after ACLR using HT-PTI might have less cartilage degeneration than those using FHT.

The purpose of this study was to investigate and compare the KOOS score and cartilage degeneration measured on MRI after ACLR in 5 years with HT-PTI and FHT, then to analyze which operation could help to slow down the cartilage degeneration after surgery and analyze the potential correlations between knee function and cartilage degeneration.

## 2. Methods

### 2.1. Participants

This study was approved by the local ethical committee and all patients signed informed consent before enrollment. This single-center, prospective, randomized trial was conducted in our hospital. The patients with ACL injury were consecutively enrolled from January to December 2014, and the indication, inclusion criteria, and other detail information were described in the methods of our previous study [24]. The inclusion criteria for participants were (1) unilateral ACL injury, (2) no history of surgery in the injured knee, and (3) age between 18 and 45 years. The participants were excluded if they had any of the following: (1) osteoarthritis; (2) combined ligament injuries; (3) multisystem trauma, nerve injuries, or fractures; or (4) cartilage injury more severe than grade 2 using the Outerbridge grading system [25] (determined during diagnostic arthroscopy) [7]. Differences of demographic data between the 2 groups were not statistically significant (all *p* values < 0.05) [7]. In all, 45 patients who qualified for inclusion were recruited and randomly distributed into 2 groups, including 21 patients underwent ACLR with HT-PTI and 24 with FHT, were performed follow-up during the periodic follow-up (Figure 1). 17.8% of the patients were lost to the follow-up.

### 2.2. Surgical Technique and Postoperative Rehabilitation

The surgical techniques had been published previously. All operations were performed by the same senior surgeon using the same instrumentation and the same arthroscopic single-bundle ACLR techniques. All patients received the same protocol of postoperative rehabilitation [1].

### 2.3. Clinical Evaluation

Considering the assessment of the outcomes during inflammatory processes and stable-state condition postoperatively [7], the evaluations were performed and documented before surgery and at 6, 12, 24, and 60 months after surgery. As our previous study summarized and published the clinical outcomes based on objective scores within 60 months postoperatively [1], the KOOS score was evaluated at pre-operation and 6, 12, 24 and 60 months post-operatively in this study. The score includes five subscales: symptoms, pain, activities of daily living (ADL), sport and recreation function (Sports/rec), and quality of life (QoL). The higher the total score, the better the outcome of knee joint after ACLR.

### 2.4. MRI Scan and Image Analysis

MRI examinations were conducted by 3.0-T MRI scanner (MAGNETOM Verio, A Tim System; Siemens, Shanghai, China) and performed at 6, 12, 24 and 60 months after ACLR. Three-dimensional double echo steady states (3D-DESS) in sagittal plane was used to quantify cartilage. The repetition time was 14.45 ms, the echo time was 5.17 ms, and the turn angle was 25°, The thickness was 1.5 mm. In sagittal T2 mapping sequence, the repetition time was 2820 ms, echo time was 13.8/27.6/41.4/55.2/69.0 ms, and turn angle was 180°, The voxel size was 0.4 × 0.4 × 3.0 mm; The visual field was 160 mm; The imaging time was 5 min 48 s. All image data were collected by Siemens software package (numaris/7, syngomr B17; Siemens) measurement and processing.

The MRI data of 3D-DESS sequence were imported into Siemens knee cap (version 1.5) workstation for automatic recognition of knee cartilage. The software could automatically divide articular cartilage into eight sub-regions [26,27]: patella (P), femoral trochlea (TrF), anterior area of lateral femoral condyle (aLFC), posterior area of lateral femoral condyle (pLFC), and anterior area of medial femoral condyle, (aMFC), posterior area of medial femoral condyle (pMFC), lateral tibia plateau (LT), medial tibia plateau (MT). The volume of cartilage in each subregion was obtained by manual fine-tuning. The aLFC and pLFC were divided by the posterior horn of the lateral meniscus, and the aMFC and pMFC were divided by the posterior horn of the medial meniscus. The specific operation interface of the software was shown in Figure 2, and the 3D model established by the software according to the preoperative knee cartilage of the patient is shown in Figure 3.

After all the MRI images being input into PACS Image processing software, two experienced radiologists independently measured the T2 value and cartilage value (CV) of each sub-region of cartilage after operation without knowing the specific grouping of FHT and HT-PTI. The repeated measurements were made on 2 days at 1–2 weeks apart [1].

The T2 values of cartilage were measured on three consecutive sagittal planes of the medial, lateral tibiofemoral and patellofemoral joints respectively (Figure 4). When manually sketching the cartilage contour for measurement, tried to avoid the subchondral bone plate and joint fluid and remove the extreme value. The average T2 value of all 3 consecutive layers was the T2 value corresponding to the measured cartilage subregion. All data of T2 mapping sequence were imported into Siemens workstation (syngi mrb17 software) for reconstruction to obtain T2 mapping.

The percent of cartilage volume changing (CV%) [17] of 8 sub-regions were measured and compared between HT-PTI and FHT groups preoperatively and at 6, 12, 24, and 60 months after ACLR. The volume change rate CV% of cartilage was calculated according to the following formula:$$\mathrm{CV}\%=\frac{\mathrm{postoperative}\;\mathrm{CV}-\mathrm{preoperative}\;\mathrm{CV}}{\mathrm{preoperative}\;\mathrm{CV}}\times 100\%$$

The negative value of CV% indicated that the cartilage volume decreased at this follow-up time point comparing to the preoperative CV, while the positive value indicated increased cartilage volume. The larger the absolute value of CV%, the greater the change of cartilage volume.

### 2.5. Statistical Analysis

Stata software (v13.0; Stata Corp., College Station, TX, USA). Continuous variables are represented by means ± standard deviation. The differences of T2 value, CV% and KOOS score between HT-PTI group and FHT group were compared. If the data obeyed normal distribution and the variance was homogeneous, the independent sample *t*-test was used; otherwise, the nonparametric Mann Whitney rank sum test was used. When comparing within groups, paired *t*-test was used if the data obeyed normal distribution and the variance was homogeneous, otherwise nonparametric Wilcoxon signed ranks test was used. Spearman correlation analysis was used to calculate and analyze the correlation between knee cartilage KOOS score, T2 and CV%. Intra correlation coefficient (ICC) was used to evaluate the consistency between the two measurements and scores (ICC < 0.4 was defined as poor; 0.4 ≤ ICC ≤ 0.75 was defined as medium; ICC > 0.75 was defined as good). The significance level was set at 0.05. Using G*Power software (version 3.1) to calculate the sample size, according to the previous relevant research to determine the corresponding research index threshold [15], set the test level α = 0.5, test efficiency (1 − β). Additionally, post hoc power analysis found that each group needs at least 16 patients to achieve significant difference. Therefore, the number of patients included in this study meets the minimum sample size requirements.

## 3. Results

Finally, 5 patients in the control group and 3 patients in the study group were lost to full follow-up. 37 participants (82.2%) undergone complete follow-ups in this study: 18 patients in the study group and 19 patients in the control group, and relevant demographic data has been published in our previous study [24]. Differences of demographic data between the two groups were not statistically significant (all *p* values > 0.05) [1].

### 3.1. Clinical Outcomes

In our previous study [24], the clinical outcomes including the International Knee Documentation Committee (IKDC), Tegner scores, Lysholm activity score, and KT-1000 arthrometer measurements were improved compared with before surgery (*p* < 0.001) and were similar in both groups.

As shown in Table 1, the scores of KOOS in the two groups showed the same trend with time, which significantly improved at 6 months (*p* < 0.001), further significantly improved at 12 months (*p* < 0.05), and then maintained at a relatively stable level from 12 months to 60 months after ACLR. The differences were statistically significant (*p* < 0.001). There was no significant difference between HT-PTI group and FHT group in the symptoms, pain, activities of daily living, sport and recreation function, and quality of life scores of KOOS at pre-operation, 6, 12, 24 and 60 months after ACLR (*p* > 0.05).

### 3.2. MRI Findings

On the MR images, no ligament re-tear or obvious cartilage defects were observed. The ICC index of inter-observer reliability was 0.786, and the ICC index of intra-observer reliability was 0.803.

### 3.3. T2 Value

The T2 mapping color scale of articular cartilage in HT-PTI group and FHT group at 6, 12, 24 and 60 months after operation is shown in Figure 5 and Figure 6. If the false color of cartilage is close to red, the cartilage had higher T2 value and more damage. Meanwhile, if the color is closer to dark blue, the cartilage had lower T2 value and less damage.

As shown in Figure 7, the preoperative cartilage T2 values had no significant difference in each sub-region of knee joint between groups (*p* > 0.05). Compared with HT-PTI group, the FHT group had higher T2 values in P, TrF, pMFC, MT and LT at 6th month, in aLFC, aMFC, MT and LT at 12 months, in TrF, aLFC, aMFC, LT and MT at 24 months, and in TrF, aLFC, aMFC, LT and MT at 60 months (all *p* < 0.05).

As shown in Table 2, in the FHT group, except from pMFC, the T2 value increased within 60 months after operation in all measured areas. In HT-PTI group, the T2 value did not change in P, TrF, pMFC and pLFC, and increased in aMFC, aLFC, MT and LT.

### 3.4. Cartilage Volume Change

In comparison with the FHT group, the HT-PTI group had similar CV% at 12 and 24 months after operation (*p* > 0.05), but significantly lower |CV%| all subregions at 6 month (all *p* < 0.05) and significantly lower |CV%| P, TrF, aMFC, aLFC, MT and LT at 60 months (all *p* < 0.05) (Figure 8).

In FHT group, the cartilage CV in all 8 sub-regions showed a transient increase at 6 months after operation, and reduced from 12 to 60th month with the increased |CV%| (*p* < 0.05). In HT-PTI group, the cartilage CV decrease with the increased |CV%| in aLFC, pMFC, pLFC, MT and LT at 24 and 60 months, and in P, TrF and aMFC at 60 months (*p* < 0.05) (Table 3).

### 3.5. Correlation Analysis

Possible associations among KOOS, T2 values and CV% are shown in Appendix A
Table A1, Table A2 and Table A3. However, no correlation was found between T2 value and KOOS scores, between CV% of each cartilage sub-region and KOOS scores, or between T2 value and CV% of each cartilage sub-region in FHT group and HT-PTI group at all timepoints (all *p* > 0.05).

## 4. Discussion

In this study, we compared the clinical KOOS score, T2 value and CV of cartilage in 5 years after ACLR with HT-PTI and FHT. The results showed that the clinical KOOS scores of all cases were significantly improved comparing to pre-operation, and there was no significant difference between the two groups. Within the 5 years after ACLR, knee cartilage injury was found in all patients, and mainly in the aMFC, aLFC, MT and LT areas. Compared to the FHT group, the cartilage damage in the HT-PTI group occurred later with the smaller area. No correlation among KOOS score, CV% and T2 values were found in all cases.

The KOOS scores of HT-PTI group and FHT group had no significant difference within 60 months after ACLR, and it had a significant improvement compared with pre-operation from the 6-month, reached the peak at the 12-month, and maintained until the 60-month after ACLR. In our previous study [24], the clinical outcomes including the IKDC, Tegner, Lysholm activity score, and KT-1000 arthrometer measurements were improved compared with before surgery (*p* < 0.001) and were similar in both groups. Different with IKDC, Tegner and Lysholm activity score, KOOS was created as a need for clinical or researching outcomes tool given to patients with cartilage injuries [19] and was designed to assess symptoms and function in younger or more active patients with ACL injuries, cartilage damage [19]. Furthermore, the KOOS has adequate internal consistency, test-retest reliability and construct validity for surgery and physical therapy after reconstruction of the ACL [19,22]. Cristiani et al. evaluated the preoperative KOOS of 73 patients undergoing ACLR with FHT for the first time, and found that the average score of the preoperative KOOS subscales were consistent with the preoperative KOOS scores of the two groups in this study. In addition, consistent with the results of this study, Macri et al. evaluated ACLR with FHT at 5 years and found that the average score of KOOS subscales were significantly improved after ACLR.

Based on T2 value on MRI, the cartilage damage in FHT group was found earlier and in more sub-regions than that in HT-PTI group in this study. Compared to pre-operation, the higher T2 values were found in aLFC and LT at 6 months in FHT group, in aMFC and MT at 12 months in FHT group, and in MT at 24 months in HT-PTI group. The similar findings were also reported in other studies of ACLR with FHT. Related studies have found that ACL injury is easy to cause contusion to the lateral tibiofemoral joint cartilage. Histologically, the proteoglycan content of the cartilage matrix in the above subregions is significantly reduced, while imageology shows that the T2 value of the cartilage in the above subregions is significantly increased [28]. It can be inferred that the cartilage degeneration in the aLFC and LT subregions occurred 6 months after the operation in this study may be due to the further aggravation of the cartilage damage in the lateral femur of the patient before the operation. Based on T2 value evaluation within one year after ACLR using FHT, Poter et al. [29] reported the risk of cartilage damage in LT sub-region was doubled, which further supported the conclusion that the cartilage of LT subregion in FHT group would degenerate in the early stage after ACLR in our study. In addition, many other related studies also found similar conclusions to this study, that the T2 value of cartilage in the medial area of tibiofemoral joint was significantly higher in the follow-up of 6-36 months than that before operation [28,30].

Regarding the CV change evaluated on MRI, the CV decreased significantly in both groups. The absolute value of CV% at 5 years in HT-PTI group was smaller than that in FHT group, which means the cartilage degenerate less in HT-PTI group. Although ACLR was to maintain knee stability and avoid cartilage damage, it had been proved that the incidence of cartilage degeneration would still high after ACLR [1]. In addition to the possible mechanical factors leading to cartilage injury after ACLR, the changes of biochemical environment in the articular cavity after reconstruction had also been proved to play a role in cartilage injury [7,8]. Among them, most of the research was the inflammation after ACL reconstruction [31,32,33,34]. Our previous studies had confirmed that ACLR with HT-PTI had less graft necrosis and less inflammation than FHT [7,23]. Therefore, HT-PTI might reduce the effect of postoperative articular cartilage by reducing necrosis and inflammation.

Interestingly, the CV% was positive at 6 months in FHT group. Relevant studies also found that compared with pre-operation, the cartilage volume increased in 3–24 months and then decreased using traditional FHT [12,13,35,36]. Wang et al. [37] conducted relevant studies and concluded that the volume increase might be caused by cartilage edema and swelling after ACLR with FHT, and they also found that there was a certain correlation between the late cartilage defect after ACLR with FHT and the early cartilage volume increase after ACLR.

Although the knee cartilage degeneration was different between two groups, both the HT-PTI and FHT groups had similar KOOS scores at all time points. The postoperative KOOS scores were significantly higher than pre-operation from the 6th month after the operation, reached the best at the 12 months and maintained until the 60 months. The correlation analysis of each group showed that there was no correlation among the KOOS score, CV% and T2 value. This might be due to the fact that the clinical score used to evaluate the prognosis was mainly based on the subjective feelings of patients [7].

## 5. Limitation

There were still some limitations in this study. Firstly, this study was limited to the quantitative monitoring of the changes of knee cartilage by MRI, but lacked of relevant clinicopathological and histological verification. However, considering the related problems of clinical ethics, it was difficult to obtain the cartilage tissue of patients after ACLR for related pathological and histological research. In addition, the current measurement of T2 value was mainly based on the measurement of cartilage T2 at multiple levels, and the average value was taken, although the measurement bias was reduced to a certain extent. However, it was also limited to the selected cartilage MRI layers, which was not the actual T2 value of the complete cartilage in some regions.

## 6. Conclusions

No matter whether FHT or HT-PTI was used for ACLR, the KOOS scores of all patients were significantly improved after their operations, and there was no significant change within 5 years after operations for both groups. The clinical outcomes of T2 and CV based on MRI confirmed that there was a certain degree of articular cartilage degeneration in both groups, and FHT group was more severe. However, there was no correlation among KOOS score, CV%, and T2 in all patients.

## Figures and Tables

**Figure 1 jcm-11-06157-f001:**
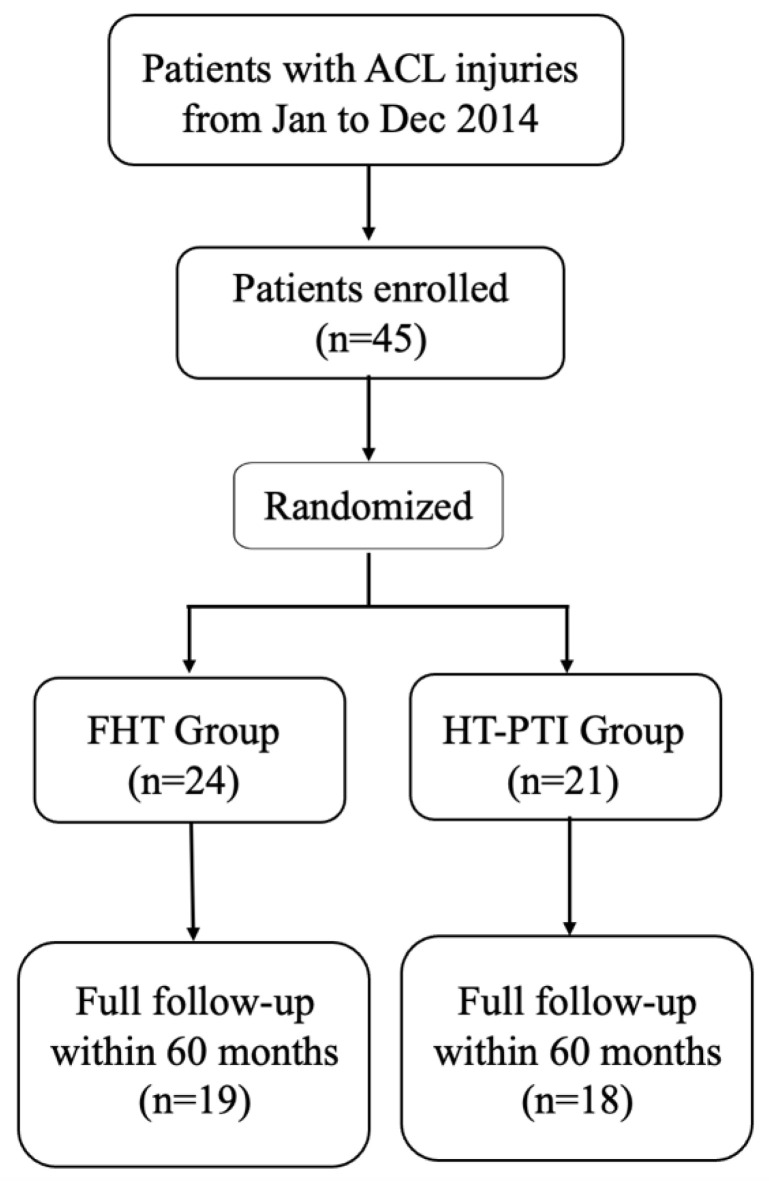
Flowchart of the randomized clinical trial. Full follow-up within 60 months included patients who had follow-up at pre-operation and 6, 12, 24, and 60 months postoperatively. The HT-PTI group had ACLR with an insertion preserved hamstring tendon autograft, and the FHT group had ACLR with a free hamstring tendon autograft. HT-PTI, hamstring tendon with intact tibial insertion; FHT, free hamstring tendon; ACL, anterior cruciate ligament; ACLR, ACL reconstruction.

**Figure 2 jcm-11-06157-f002:**
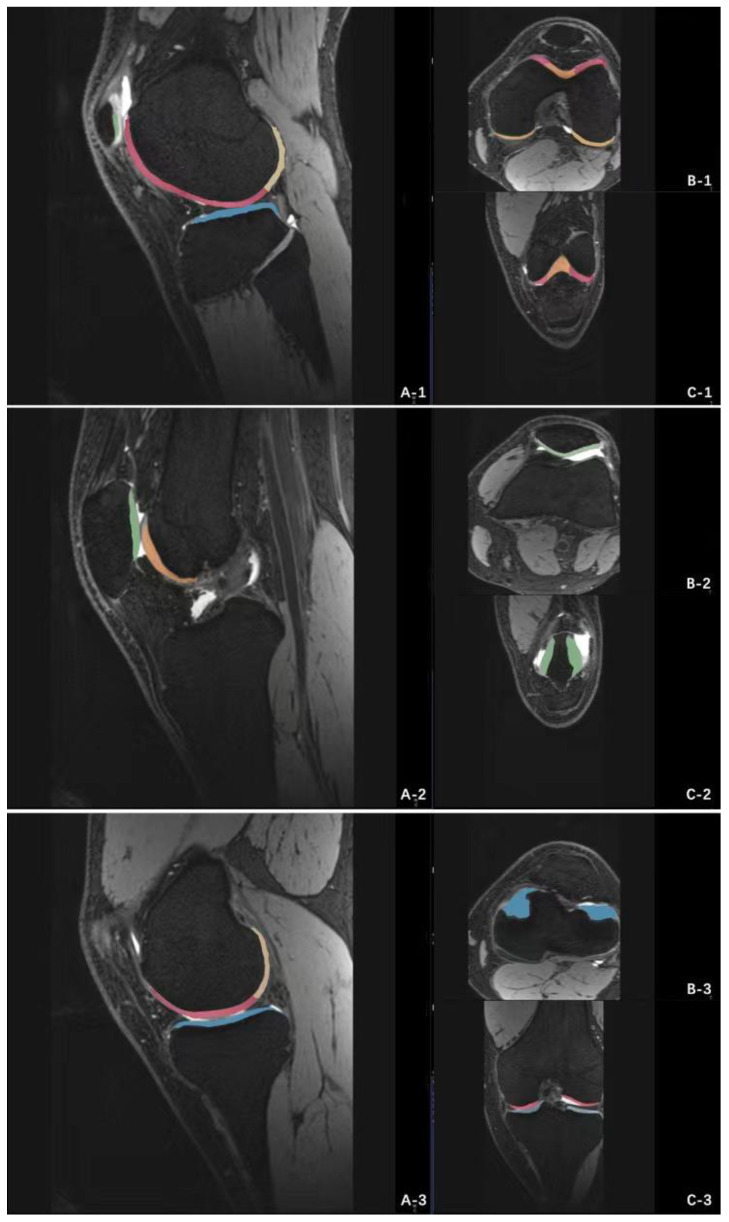
Siemens knee cap (version 1.5) software 3D-DESS image workstation operation interface. The software can automatically recognize and calculate the volume of each cartilage subregion of the knee joint. (**A-1**), Sagittal position of lateral knee joint; (**B-1**), Horizontal position of lateral knee joint; (**C-1**), Coronal position of lateral knee joint; (**A-2**), Sagittal position of middle knee joint; (**B-2**), Horizontal position of middle knee joint; (**C-2**), Coronal position of middle knee joint; (**A-3**), Sagittal position of medial knee joint; (**B-3**), Horizontal position of medial knee joint; (**C-3**), Coronal position of medial knee joint.

**Figure 3 jcm-11-06157-f003:**
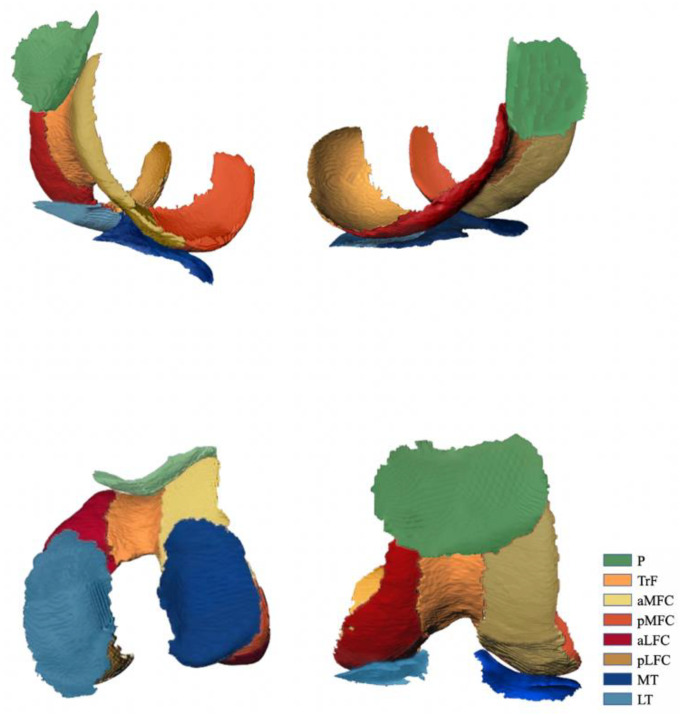
3D model of knee joint cartilage. The 3D reconstruction model of complete knee cartilage was automatically divided into 8 subregions: P, TrF, aMFC, pMFC, aLFC, pLFC, MT and LT by Siemens knee cap (version 1.5). Different cartilage subareas are marked with different colors. The corresponding cartilage subareas of each color are shown in the far right of this figure. P, patella. TrF, femoral trochlea. aLFC, anterior area of lateral femoral condyle. pLFC, posterior area of lateral femoral condyle. aMFC, anterior area of medial femoral condyle. pMFC, posterior area of medial femoral condyle. LT, lateral tibia plateau. MT, medial tibia plateau.

**Figure 4 jcm-11-06157-f004:**
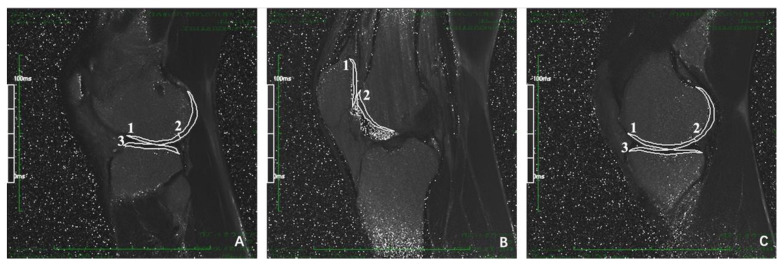
Measurements of T2 value of cartilage in each sub region of knee joint. (**A**) the lateral tibiofemoral joint of the knee: the measurement sub zone 1 is the aLFC, the measurement sub zone 2 is the pLFC, and the measurement sub zone 3 is the LT; (**B**) patellofemoral joint of knee joint: the measurement sub zone 1 is P, and the measurement sub zone 2 is TrF; (**C**) medial tibiofemoral joint of knee joint: the measurement subzone 1 is aMFC, the measurement subzone 2 is pMFC, and the measurement subzone 3 is MT.

**Figure 5 jcm-11-06157-f005:**
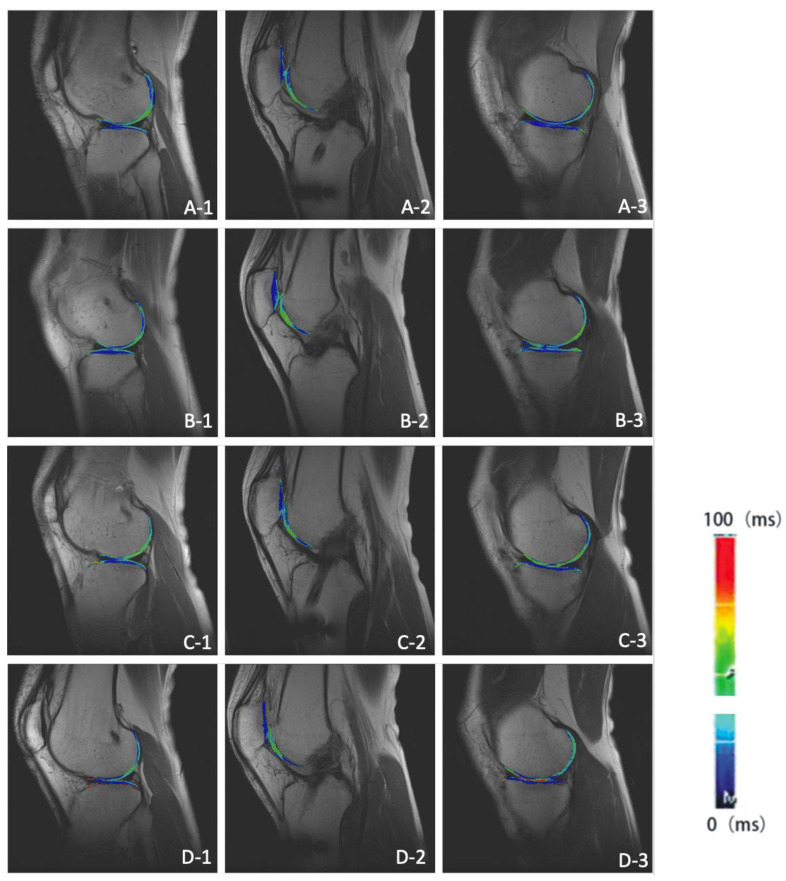
Sagittal T2 mapping of knee joint in HT-PTI group. (**A-1**–**A-3**) show the lateral tibiofemoral joint, patellofemoral joint and medial tibiofemoral joint in HT-PTI group at 6 months after operation; (**B-1**–**B-3**) show the lateral tibiofemoral joint, patellofemoral joint and medial tibiofemoral joint in HT-PTI group 12 months after operation; (**C-1**–**C-3**) show the lateral tibiofemoral joint, patellofemoral joint and medial tibiofemoral joint in HT-PTI group 24 months after operation; (**D-1**–**D-3**) show the lateral tibiofemoral joint, patellofemoral joint and medial tibiofemoral joint in HT-PTI group 60 months after operation.

**Figure 6 jcm-11-06157-f006:**
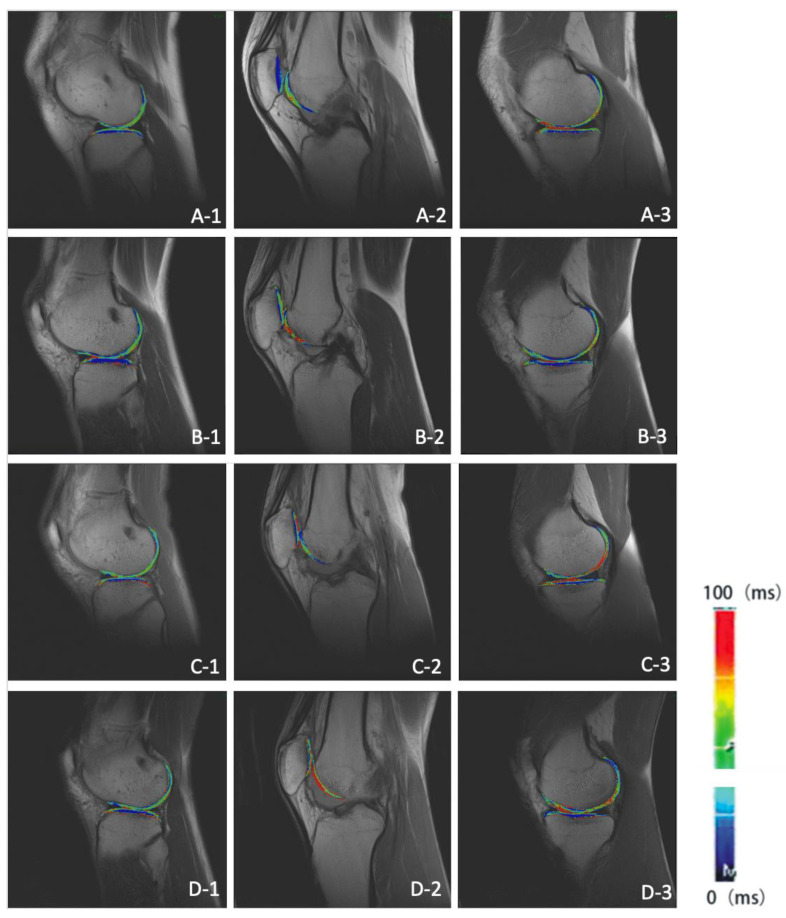
T2 mapping of knee joint sagittal plane in FHT group. (**A-1**–**A-3**) show the lateral tibiofemoral joint, patellofemoral joint and medial tibiofemoral joint at 6 months after operation in FHT group; (**B-1**–**B-3**) show the lateral tibiofemoral joint, patellofemoral joint and medial tibiofemoral joint 12 months after operation in FHT group; (**C-1**–**C-3**) show the lateral tibiofemoral joint, patellofemoral joint and medial tibiofemoral joint 24 months after operation in FHT group; (**D-1**–**D-3**) show the lateral tibiofemoral joint, patellofemoral joint and medial tibiofemoral joint in FHT group 60 months after operation.

**Figure 7 jcm-11-06157-f007:**
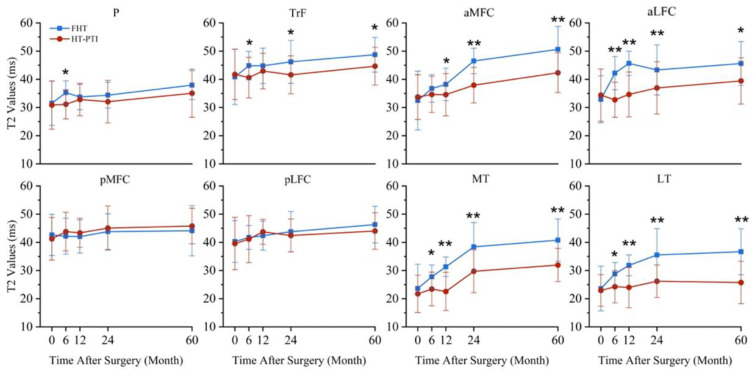
The T2 values of cartilage in each subregion were compared between FHT group and HT-PTI group. There was no significant difference in T2 value between the two groups before operation; The difference of P T2 value between two groups was significant at 6 months after operation; The differences of TrF T2 values between two groups were significant at 6, 24 and 60 months after operation; The differences of aMFC T2 values between two groups were significant at 12, 24 and 60 months after operation; The T2 values of aLFC, MT and LT between two groups were significantly different within 60 months after operation. * *p* < 0.05; ** *p* < 0.001. FHT, free hamstring tendon; HT-PTI, hamstring tendon with intact tibial insertion; P, patella; TrF, femoral trochlea; aLFC, anterior area of lateral femoral condyle; pLFC, posterior area of lateral femoral condyle; aMFC, anterior area of medial femoral condyle; pMFC, posterior area of medial femoral condyle; LT, lateral tibia plateau; MT; medial tibia plateau.

**Figure 8 jcm-11-06157-f008:**
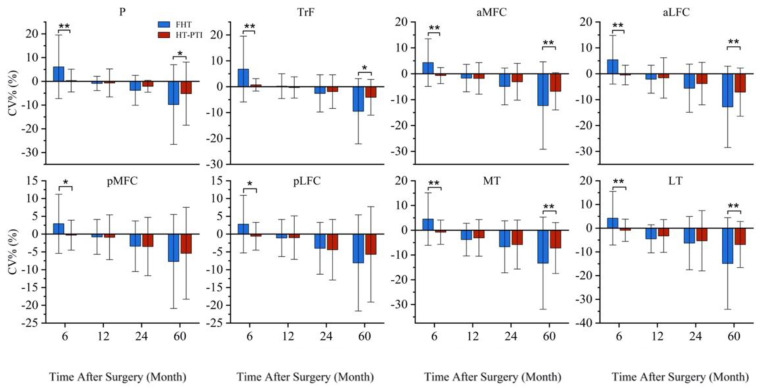
The CV% of cartilage between FHT group and HT-PTI group were compared. At 6 months after operation, significant differences were observed in all subgroups; Significant differences were observed in P, TrF, aMFC, aLFC, MT and LT at 60 months between two groups after operation. * *p* < 0.05; ** *p* < 0.001. CV%, the percent of cartilage volume changing; FHT, free hamstring tendon; HT-PTI, hamstring tendon with intact tibial insertion; P, patella; TrF, femoral trochlea; aLFC, anterior area of lateral femoral condyle; pLFC, posterior area of lateral femoral condyle; aMFC, anterior area of medial femoral condyle; pMFC, posterior area of medial femoral condyle; LT, lateral tibia plateau; MT; medial tibia plateau.

**Table 1 jcm-11-06157-t001:** KOOS Outcomes in the Study and Control Groups ^a^.

	Symptoms	Pain	ADL	Sports/Rec	QoL
Control Group					
Pre-operative	67.1 ± 22.3	75.2 ± 23.8	80.8 ± 21.7	51.3 ± 18.9	47.3 ± 24.9
6-month	82.5 ± 15.3 *	84.9 ± 13.8 *	89.9 ± 11.3 *	63.3 ± 21.2 *	81.6 ± 17.3 *
12-month	90.2 ± 11.7 *^#^	91.0 ± 9.9 *^#^	95.6 ± 5.5 *^#^	80.4 ± 8.7 *^#^	92.4 ± 7.2 *^#^
24-month	92.1 ± 13.9 *^#^	93.7 ± 7.4 *^#^	97.1 ± 4.2 *^#^	86.9 ± 10.6 *^#^	93.8 ± 6.1 *^#^
60-month	92.3 ± 10.6 *^#^	93.3 ± 6.3 *^#^	96.7 ± 4.4 *^#^	86.3 ± 11.1 *^#^	92.1 ± 5.8 *^#^
Study group					
Pre-operative	68.9 ± 23.7	73.7 ± 21.1	79.8 ± 22.5	53.7 ± 19.6	50.1 ± 22.8
6-month	83.1 ± 15.9 *	86.3 ± 12.6 *	88.5 ± 10.9 *	65.1 ± 19.7 *	80.8 ± 15.9 *
12-month	89.9 ± 10.8 *^#^	92.2 ± 13.3 *^#^	96.8 ± 5.3 *^#^	81.2 ± 10.4 *^#^	91.7 ± 9.9 *^#^
24-month	91.9 ± 12.5 *^#^	94.5 ± 7.1 *^#^	98.1 ± 4.7 *^#^	87.5 ± 12.0 *^#^	93.2 ± 7.7 *^#^
60-month	92.7 ± 11.2 *^#^	93.6 ± 7.7 *^#^	97.3 ± 4.5 *^#^	85.6 ± 11.5 *^#^	92.5 ± 6.2 *^#^

^a^ Clinical-outcomes in the study and control groups at pre-operation, 6, 12, 24 and 60 months after ACLR. KOOS: Knee Injury and Osteoarthritis Outcome Score, ADL: activities of daily living, Sports/Rec: sport and recreation function, QoL: quality of life. Comparing with the pre-operative clinical outcome, * *p* < 0.001. Comparing with the 6-month clinical outcome, ^#^
*p* < 0.05. Values were shown as mean ± SD.

**Table 2 jcm-11-06157-t002:** T2 values at Different Subregions and Timepoints in Two Groups ^α^.

	P	TrF	aMFC	aLFC	pMFC	pLFC	MT	LT
FHT group								
Pre-operative	31.52 ± 7.84	40.85 ± 9.77	32.46 ± 10.46	32.89 ± 8.35	42.63 ± 7.32	40.28 ± 7.39	23.63 ± 8.57	23.58 ± 7.92
6 m	35.28 ± 4.16	44.83 ± 5.04	36.73 ± 4.88	42.19 ± 5.89 ^a^	42.18 ± 6.37	41.73 ± 4.22	27.78 ± 4.20	28.82 ± 4.01 ^a^
12 m	33.72 ± 4.49	44.79 ± 6.27	38.18 ± 5.75 ^a^	45.64 ± 4.28 ^a^	42.04 ± 5.86	42.35 ± 5.14	31.32 ± 3.48 ^a^	31.83 ± 3.69 ^a^
24 m	34.38 ± 4.56	46.21 ± 7.61 ^a^	46.51 ± 4.52 ^abc^	43.33 ± 8.87 ^a^	43.81 ± 6.27	43.77 ± 7.14	38.42 ± 8.59 ^abc^	35.53 ± 9.31 ^ab^
60 m	37.93 ± 5.12 ^a^	48.73 ± 6.16 ^a^	50.62 ± 8.18 ^abcd^	45.61 ± 7.79 ^a^	44.11 ± 8.92	46.29 ± 6.51 ^ab^	40.78 ± 7.47 ^abc^	36.67 ± 8.15 ^ab^
HT-PTI group								
Pre-operative	30.86 ± 8.56	41.75 ± 8.97	33.72 ± 7.93	34.38 ± 9.28	41.24 ± 7.53	39.56 ± 9.31	21.74 ± 6.62	22.94 ± 5.61
6 m	31.16 ± 5.18	40.63 ± 7.21	34.63 ± 6.39	32.74 ± 6.23	43.81 ± 6.89	41.13 ± 8.35	23.42 ± 5.96	24.27 ± 5.75
12 m	32.84 ± 5.73	42.94 ± 6.33	34.56 ± 7.49	34,62 ± 7.97	43.38 ± 5.12	43.74 ± 4.42	22.56 ± 6.77	24.01 ± 7.20
24 m	32.06 ± 7.58	41.60 ± 6.73	37.92 ± 6.32	36.96 ± 9.27	45.06 ± 7.85	42.42 ± 5.84	29.74 ± 7.62 ^abc^	26.19 ± 5.75
60 m	35.05 ± 8.53	44.65 ± 6.69	42.34 ± 7.05 ^abcd^	39.43 ± 8.21 ^b^	45.79 ± 6.29	44.02 ± 6.47	31.94 ± 5.87 ^abc^	27.75 ± 7.53 ^a^

^α^ Values were shown as mean ± SD. ^a^ Comparing with the pre-operative T2-values, *p* < 0.05. ^b^ Comparing with the 6-month-T2-values, *p* < 0.05. ^c^ Comparing with the 12-month-T2-values, *p* < 0.05. ^d^ Comparing with the 24-month-T2-values, *p* < 0.05. FHT, free hamstring tendon; HT-PTI, hamstring tendon with intact tibial insertion; P, patella; TrF, femoral trochlea; aLFC, anterior area of lateral femoral condyle; pLFC, posterior area of lateral femoral condyle; aMFC, anterior area of medial femoral condyle; pMFC, posterior area of medial femoral condyle; LT, lateral tibia plateau; MT; medial tibia plateau.

**Table 3 jcm-11-06157-t003:** CV% at Different Subregions and Timepoints in Two Groups ^α^.

	P	TrF	aMFC	aLFC	pMFC	pLFC	MT	lt
FHT group								
6 m	6.1 ± 13.4	6.8 ± 12.7	4.3 ± 9.2	5.4 ± 9.4	2.9 ± 8.3	2.8 ± 8.1	4.5 ± 10.6	4.2 ± 11.3
12 m	−0.9 ± 3.0 *	0.2 ± 4.8 *	−1.7 ± 5.3 *	−2.1 ± 5.4 *	−0.8 ± 4.9 *	−1.1 ± 5.2 *	−3.8 ± 6.6 *	−4.5 ± 5.9 *
24 m	−3.8 ± 6.3 *^#^	−2.6 ± 7.2 *^#^	−4.9 ± 7.1 *^#^	−5.6 ± 9.3 *^#^	−3.4 ± 7.1 *^#^	−4.0 ± 7.3 *^#^	−6.7 ± 10.5 *^#^	−6.3 ± 11.2 *^#^
60 m	−9.8 ± 16.8 *^#+^	−9.5 ± 12.6 *^#+^	−12.3 ± 16.9 *^#+^	−12.8 ± 15.7 *^#+^	−7.7 ± 13.2 *^#+^	−8.1 ± 13.5 *^#+^	−13.3 ± 18.6 *^#+^	−14.9 ± 19.3 *^#+^
HT-PTI group								
6 m	0.3 ± 4.8	0.7 ± 2.4	−0.7 ± 3.1	−0.5 ± 3.8	−0.3 ± 4.2	−0.6 ± 3.9	−0.8 ± 4.9	−0.9 ± 4.7
12 m	−0.7 ± 5.9	−0.3 ± 4.1	−1.8 ± 6.1	−1.6 ± 7.8	−0.9 ± 6.3	−1.0 ± 6.1	−3.1 ± 7.4	−3.3 ± 6.9
24 m	−2.1 ± 2.5	−1.9 ± 6.5	−3.1 ± 7.1	−3.8 ± 8.2 *	−3.5 ± 8.2 *	−4.4 ± 8.5 *	−5.8 ± 9.9 *	−5.3 ± 12.7 *
60 m	−5.2 ± 13.3 *^#+^	−4.1 ± 6.9 *^#^	−6.8 ± 7.2 *^#+^	−7.1 ± 9.3 *^#+^	−5.4 ± 12.9 *^#^	−5.7 ± 13.4 *^#^	−7.2 ± 10.3 *^#^	−6.9 ± 9.7 *^#^

^α^ Values were presented as mean ± SD. * Comparing with the 6 m-CV%, *p* < 0.05; ^#^ Comparing with the 12 m-CV%, *p* < 0.05; ^+^ Comparing with the 24 m-CV%, *p* < 0.05. CV%, the percent of cartilage volume changing; FHT, free hamstring tendon; HT-PTI, hamstring tendon with intact tibial insertion; P, patella; TrF, femoral trochlea; aLFC, anterior area of lateral femoral condyle; pLFC, posterior area of lateral femoral condyle; aMFC, anterior area of medial femoral condyle; pMFC, posterior area of medial femoral condyle; LT, lateral tibia plateau; MT; medial tibia plateau.

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
