# Peer review of "Knee Cartilage Change within 5 Years after Aclr Using Hamstring Tendons with Preserved Tibial-Insertion: A Prospective Randomized Controlled Study Based on Magnetic Resonance Imaging"

_jcm, 2022, doi:10.3390/jcm11206157_

Round 1
Reviewer 1 Report
The study deals with the assessment of the knee outcomes – KOOS and MRI T2 relaxation time of the articular cartilage, both evaluated at different follow-up, i.e., pre-operative, and 6, 12, 24 and 60 months post-operative – after ACL reconstruction provided by a single-bundle technique but considering two different approaches, i.e., with and without preserving the tibial insertion of the hamstring tendon, thus to possibly ensure blood supply of the graft. The authors finding suggests that no difference occurred in terms of KOOS between the investigated groups, despite an improvement of such outcome is reached compared to pre-operative condition. Moreover, authors reported also that cartilage degeneration results delayed by preserving the tibial insertion of the hamstring tendon.
Despite the topic is relevant and of great interest, some minor concerns should be reported.
Regarding Introduction, it surely provides sufficient background, but it is not clearly reported which is the main focus of the study, i.e., comparing the outcomes of two different approaches in reconstructing a torn ACL. In this perspective, authors should consider to re-write partially this section, better highlighting that the major concern and the driven idea at the base of the study regard sub-optimal techniques to date applied to the reconstruction on ACL.
Regarding Methods, the authors should consider to report the demographic data of the patients involved and, moreover, the inclusion criteria, despite such information were highlighted in a previous study.
Authors should consider to revise Figure 1, aiming to better evidence the choice of the different follow-up; in this perspective, why the authors choose such specific time-points of evaluation? Of course the firsts, i.e., 6, 12 and 24 months, involved the assessment of the outcomes during inflammatory and post-operative processes, while 60 months should be ideally considered as a time-point in which the reconstructed knee reached a stable-state condition. Nevertheless, such information should be reported in the manuscript, thus to support the authors choice, also reporting relative literature.
Regarding MRI evaluations, authors should consider to add more details about their analysis. In this perspective, which algorithm was used to obtain T2 maps from the T2 clinical sequences? Moreover, in evaluating knee cartilage T2, possible values of T2 are generally limited to a specific amount, thus to specifically avoid of intra-articular fluids, especially in the pre-operative assessment. Did authors consider to properly trimmer the extent of T2 values attributable to knee articular cartilage? Regarding the ROIs evaluated, authors should consider to report why they focus on such specific articular areas.
Authors should double-check in the manuscript the use of CV and CV%, thus to avoid misunderstandings.
Authors should consider to investigate not only absolute values of the investigated parameters, but also their differences during the follow-up – e.g., as reported considering cartilage volume – which could represent a more informative manner to highlight temporal trend peculiar of clinical outcomes of knee joint after ACL reconstruction.
Regarding Results, authors should consider to revise Figure 5 and Figure 6, by adding MR slices of the investigated compartment in the pre-operative phase.
The major concern about the study regards Discussion. Despite authors compared their findings to the relative literature, no suggestions are reported considering how the different approaches investigated, i.e., reconstructing ACL with or without preservation of the tibial-insertion, impact on clinical outcomes. Authors must properly argue about their findings. Moreover, authors should consider to take into account how the graft peculiarities, i.e., maturation, possibly affect knee cartilage status. In this perspective, studies reported how graft maturation impacts on knee joint, considering both functional and structural/biological outcomes (for example, see the following studies: 1. https://doi.org/10.1002/jor.24572 2. https://doi.org/10.3390/life11121383 3. https://doi.org/10.1186/s13018-019-1193-y). Accordingly, authors should consider to investigate also graft main peculiarities, e.g., change in volume and T2 relaxation, through the follow-up.
Considering the reported information, the manuscript will reach a completeness to be published in Journal of Clinical Medicine only if the above reported issues and observation will be responded.
Reviewer 2 Report
Dear Authors, I think the study is very interesting.
The article carefully compares the anterior cruciate ligament reconstructions using free Hamstring tendons with preserved tibial-insertion Hamstring tendon.
The paper is very interesting and relevant in its field.
However, some minor revisions are needed.
- The manuscript is written correctly in English, although some grammatical errors should be corrected.
- Title
I suggest you create a more provocative and direct title, including the main findings of your study
- Limitation
Almost 18% of the patients were lost to the follow-up. Add this information in the limitation.
I suggest you include an interesting manuscript regarding the topic
- Corona K, Cerciello S, Vasso M, et al. Age over 50 does not predict results in anterior cruciate ligament reconstruction. Orthopedic Reviews. 2022;14(5). doi:10.52965/001c.37310
Reviewer 3 Report
Interesting Topic and well-written manuscript.
Title: interesting
Introduction: well-written and not too long
Methods: in detail described, very good illustration with figures and tables
Results: ok! The onl ything waht is missing, is the clear and direct advantage of this surgical technique, the blood supply of the graft and the integration of the graft. Here is nearly nothing described about the status (MRI evaluation, clinical stability like lachman, etc.).
Discussion, limitation and conlusion: ok. Only the direct influence form the surgical technique to cartilage results in the same knee shopuld be better descibed in introduction and discussion
